Pharmacoscreening, molecular dynamics, and quantum mechanics of inermin from Panax ginseng: a crucial molecule inhibiting exosomal protein target associated with coronary artery disease progression

V Janakiraman 1
M Sudhan 1
Wani Abubakar 2
Ahmad Sheikh F. 3
Nadeem Ahmed 3
Sharma Ashutosh 4
Ahmed Shiek S. S. J. shiekssjahmed@gmail.com 1
1 Muti-omics and Drug Discovery Lab, Faculty of Allied Health Sciences, Chettinad Hospital and Research Institute, Chettnad Academy of Research and Education , Kelambakkam , Tamil Nadu , India
2 Department of Immunology, St. Jude Children’s Research Hospital Memphis , TN , USA
3 Department of Pharmacology and Toxicology, College of Pharmacy, King Saud University , Riyadh , Saudi Arabia
4 Centre of Bioengineering, NatProLab, Plant Innovation Lab, School of Engineering and Sciences, Tecnologico de Monterrey , Queretaro , Mexico
Gokara Mahesh
Electronic publication date: 2023 Dec 6
Publication date: 2023
Volume: 11
Electronic Location ID: e16481
Received 2023 Jun 21; Accepted 2023 Oct 26
Copyright: ©2023 Janakiraman et al.
Copyright year: 2023
Copyright holder: Janakiraman et al.
License: This is an open access article distributed under the terms of the Creative Commons Attribution License, which permits unrestricted use, distribution, reproduction and adaptation in any medium and for any purpose provided that it is properly attributed. For attribution, the original author(s), title, publication source (PeerJ) and either DOI or URL of the article must be cited.
License URL: https://creativecommons.org/licenses/by/4.0/

Keywords: Exosomes, Systems biology, Protein network, Density functional theory, Molecular dynamic simulation, Coronary artery disease

Funding: Researchers Supporting Project King Saud University, Riyadh, Saudi Arabia RSPD2023R709 This research was funded by King Saud University, Riyadh, Saudi Arabia, Project Number (RSPD2023R709). The funders had no role in study design, data collection and analysis, decision to publish, or preparation of the manuscript.

==============================
Background

Exosomes, microvesicles, carry and release several vital molecules across cells, tissues, and organs. Epicardial adipose tissue exosomes are critical in the development and progression of coronary artery disease (CAD). It is hypothesized that exosomes may transport causative molecules from inflamed tissue and deliver to the target tissue and progress CAD. Thus, identifying and inhibiting the CAD-associated proteins that are being transported to other cells via exosomes will help slow the progression of CAD.

Methods

This study uses a systems biological approach that integrates differential gene expression in the CAD, exosomal cargo assessment, protein network construction, and functional enrichment to identify the crucial exosomal cargo protein target. Meanwhile, absorption, distribution, metabolism, and excretion (ADME) screening of Panax ginseng-derived compounds was conducted and then docked against the protein target to identify potential inhibitors and then subjected to molecular dynamics simulation (MDS) to understand the behavior of the protein-ligand complex till 100 nanoseconds. Finally, density functional theory (DFT) calculation was performed on the ligand with the highest affinity with the target.

Results

Through the systems biological approach, Mothers against decapentaplegic homolog 2 protein (SMAD2) was determined as a potential target that linked with PI3K-Akt signaling, Ubiquitin mediated proteolysis, and the focal adhesion pathway. Further, screening of 190 Panax ginseng compounds, 27 showed drug-likeness properties. Inermin, a phytochemical showed good docking with −5.02 kcal/mol and achieved stability confirmation with SMAD2 based on MDS when compared to the known CAD drugs. Additionally, DFT analysis of inermin showed high chemical activity that significantly contributes to effective target binding. Overall, our computational study suggests that inermin could act against SMAD2 and may aid in the management of CAD.

Introduction

Coronary artery disease (CAD) is one of the most prevalent diseases that contributes to high mortality worldwide (Faccini et al., 2017) with a significant impact on social and economic loss in both developed and developing countries (Afiqah Hamzah et al., 2021). According to a WHO report, the global burden of cardiovascular disease accounts for 32% deaths each year (World Health Organization, 2021). CAD causes coronary artery stenosis, which leads to hypoxia, ischemia, and necrosis (Zhang et al., 2020). Several risk factors are claimed to cause CAD, including age, obesity, smoking, family history, diabetes, heredity, and dyslipidemia (Iyengar et al., 2017). At the molecular level, CAD presents as the accumulation of lipid species, apoptosis, inflammation, and oxidative damage in arteries; and their effects were observed in various tissues including blood (Zaman et al., 2000). Current CAD management is primarily focused on anti-thrombotic, anti-inflammatory, antioxidant, and lipid-lowering drugs. In extreme or myocardial infarction conditions, surgical procedures are recommended to benefit individuals with CAD (Eikelboom et al., 2021). Recently, exosomes were observed to play a major role in regulating the cell signaling pathway (Nguyen, Azam & Wang, 2021). In particular, exosomes have been shown to transport causative molecules such as inflammatory, oxidative, and lipid species across the cells/tissues that progress CAD (Zhang et al., 2021; Wang et al., 2019). Henceforth, studies recommend targeting exosomal cargo CAD molecules that may help to inhibit or delay the progression of disease.

Exosomes are microvesicles sized between ∼20–140 nm in diameter (Théry, Zitvogel & Amigorena, 2002). Exosomes are produced by almost every cell of the human body (Li et al., 2017) that circulate along with the body fluids as a part of regular cellular communication (Chung et al., 2020; Sun et al., 2018). Apart from regular molecular transport, exosomes carry over-flooded causative molecules and debris from one cell to another that protect cells from cellular stress (Han et al., 2022). In our previous research, we showed that exosomes transport viral components as well as the over-flooded inflammatory molecules from the lungs to the brain causing neuronal damage during COVID infection (Ahmed et al., 2021). Similarly in tumor conditions, the exosome carries vital components from the primary tumor region to other organs causing metastasis. In addition to these, exosomes are reported to play a vital role in the transport of pathological components in diseases such as infectious diseases, neurological diseases, and CAD (Feng et al., 2019). Interestingly, the pathogenic role of exosomes in CAD was reported to be involved in disease progression (Li et al., 2022). Therefore, inhibiting the CAD-associated proteins that are being transported to other cells/tissue via exosomes will help to slow the progression of CAD.

Recently, traditional Chinese medicine has been widely used by Eastern Asians and also grabbed the attention of many researchers for its effective treatment of various diseases (Xue et al., 2012). Studies have confirmed the benefit of phytochemicals from Chinese plants for the treatment of CAD (Hao et al., 2017). Of various Chinese plants, one of the most prominently used plant products in China is Asian ginseng (Panax ginseng C.A. Meyer). The beneficial effects of Panax ginseng include inhibiting oxidation, improving blood circulation, and protecting the cerebrovascular system (Kim, 2018). Sathishkumar et al. (2012) report that the bioactive ingredient of Panax ginseng includes polyacetylene alcohols, peptides, fatty acids, polysaccharides, and ginsenosides (Sathishkumar et al., 2012). Of these, ginsenosides (triterpene saponins) is one of the major active components of Panax ginseng proven to have various pharmacological activities, such as anti-inflammatory, anti-oxidant, anti-hypertensive, anti-atherosclerotic, immunomodulatory, anti-hypertensive, anti-hyperglycemic, anti-stress, and anti-cancer (Sathishkumar et al., 2012). Exploring the pharmacological effect of Panax ginsengderived compounds and their relevance in inhibiting the CAD proteins being transported via exosome will be promising for therapeutic application.

Herein, using a systems biological approach, we identified the crucial CAD protein that can be transported via exosome-based on the network analysis. Then, we primarily screened natural compounds of the Panax ginseng plant and based the screening on absorption, distribution, metabolism, and excretion (ADME) properties. Further, the affinity of ADME-screened compounds to the CAD target protein was assessed based on molecular docking. Furthermore, the molecular dynamics simulation (MDS) was used to investigate the structural conformation of the target protein with the docked compound. Additionally, the quantum property of the selected compound was studied through density functional theory (DFT) analysis. Overall, our study establishes the potential of a natural compound that could be useful in the CAD treatment.

Materials & Methods

Collection and optimization of natural compounds

The bioactive compounds of Panax ginseng were identified and its structures were obtained from the Traditional Chinese Medicine Systems Pharmacology (TCMSP) database (https://tcmsp-e.com/tcmsp.php). Then, the retrieved compound structures were optimized with a OPLS3force-field using the LigPrep tool in the Schrödinger suite (Maestro 11.2). Likewise, the known CAD drugs (amlodipine, furosemide, digoxin, clopidogrel, and atorvastatin) were downloaded as structure data files (SDF) format from PubChem (https://pubchem.ncbi.nlm.nih.gov/) for comparative analysis. Then the optimized natural compounds were screened with QikProp (Maestro, Schrödinger suite) (Ntie-Kang et al., 2013) to predict their absorption, distribution, metabolism and excretion (ADME) property based on molecular descriptors such as molecular weight, donor hydrogen bond, acceptor hydrogen bond, water partition coefficient, Caco-2 cell permeability, central nervous system (CNS) activity, blood/brain partition coefficient, cell permeability, oral absorption, serum albumin binding, Lipinski rule of five, and Jorgensen’s rule (Ntie-Kang et al., 2013; Rajagopal et al., 2020).

CAD genes and exosome

A gene expression array dataset from the Gene Expression Omnibus (GEO) (https://www.ncbi.nlm.nih.gov/geo/) was selected with the following criteria that includes, (1) dataset containing expression profile of more than three samples in a group, (2) dataset containing CAD participants with the appropriate control executed in the CAD sample from the cardiac associated tissue, (3) the dataset providing human gene expression profile that uses microarray or RNAseq technique was considered, and (4) experiment carried out in animal or tissue culture were eliminated/excluded. To the selected dataset, limma R-program was implemented to identify the differentially expressed genes (DEGs) in CAD with log2fold-change (FC) and p-value < 0.05. Among DEGs, only the over expressed genes were selected for further assessment.

Intersection of DEGs with exosomal proteins cargo

To gather the information on exosomal cargo proteins, the EXOCARTA database was used (http://www.exocarta.org/) which contains the exosomal cargo of DNA, mRNA, miRNA, lipids, and proteins from multiple species. All human exosomal cargo proteins were downloaded and duplicates were removed to obtain an unique protein list. These exosomal unique lists were mapped to the over expressed DEGs of CAD to discover a common protein list (SET-A). The proteins in SET-A were demonstrated to have qualities (1) over expressed in CAD and (2) carried via exosome. Then the SET-A was used to build an inter-protein network with the STRING database (https://string-db.org/). Thereby, the constructed network contains both SET-A and its interacting proteins, resembling that SET-A on exosomal delivery into a target cell can readily interact and activate the cellular proteins to perform functions.

Enrichment analysis

The function of these network proteins was assessed by gene ontology (GO) and KEGG pathway using ShinyG0 0.77 (http://bioinformatics.sdstate.edu/go/). The GO analysis was carried out to identify the biological process, molecular function, and cellular component of the network proteins with a false discovery rate (FDR) less than 0.05. Similarly, the molecular pathways analysis were explored using KEGG pathway module (FDR < 0.05).

Molecular docking

From the protein interaction network, the SET-A proteins with a high degree of interaction were selected as target and their 3D protein structure was retrieved from the Protein Data Bank (PDB) database. Using a protein preparation module (Schrödinger suite), the target protein was optimized, which removes water molecules, heteroatoms, ions, and unwanted protein chains. Similarly, the LigPrep module (Schrödinger suite) was used for compound preparation such as adding hydrogen bonds. The binding pockets of protein were identified by P2Rank tool (https://prankweb.cz/) for grid generation around the pockets (Umashankar et al., 2021). Simultaneously, the OPLS3 force-field was applied to protein and small molecules (natural compounds + CAD drugs). A Glide suite (Schrödinger suite) was used to perform docking, to find the molecular interaction between the protein with natural compounds and also with the known CAD drugs. Finally, the small molecules with least binding energy was considered for molecular dynamic simulation.

Molecular dynamics simulation

After gaining the understanding of the ligand–protein complex’s through docking, the molecular dynamics (MD) simulation was performed for the target protein with the compound from Panax ginseng and CAD drug having highest binding affinity. For MD simulations, the Desmond tool included in the Schrödinger Drug Design Suite was utilized. To investigate the stability of the protein-ligands complex was carried out for 100 nanosecond (ns). Each simulation was carried out in three phases involving system construction, minimization, and the simulation run. Following the determination of the docked ligand–protein complex, the system was modeled using the SPC solvent system within the orthorhombic boundary and ions neutralized then subjected to energy minimization until it reached a gradient threshold of 25 kcal/mol at a pressure of 1 bar and a temperature of 300 K using the NPT ensemble class. The OPLS2005 force field optimization was used to minimize the system by carrying 2,000 iterations (Convergence scale of 1 kcal/mol/Å). A Nose–Hoover Chain thermostat was used to maintain the temperature, while a Martyna-Tobias-Klein barostat was utilized to sustain pressure with the equilibrium was set 200 ps. Finally, root-mean-square-deviation (RMSD), RMSF root-mean-square-fluctuation (RMSF) trajectories of the protein and ligand, protein–ligand interactions, and contacts with a variety of amino acids were used to determine the stability of the complex during the MD simulation.

Quantum computation calculation

The DFT calculation was preceded by using the Jaguar suite in Schrödinger (Yele et al., 2021). The predictions of chemical compound properties were calculated with the help of the integrated function of the B3LYP technique along with 6-311G** ++ basis set (Yele et al., 2021). The geometrical parameters have been calculated theoretically including bond angle, bond length, and dihedral angles (Deghady et al., 2021). Similarly, highest occupied molecular orbital (HOMO) and lowest unoccupied molecular orbital (LUMO) values were utilized to calculate other parameters (such as chemical potential, optical softness, chemical hardness, net electrophilicity, electrophilicity index, nucleophilicity index, and other quantum chemical parameters) by using Formulas 1–8. Likewise, the B3LYP/6-311++G (d,p) method in DFT was employed to identify the molecular electrostatic potential (ESP) and Mulliken charge analysis of the lead compound (Deghady et al., 2021).

(1) ΔE=ELUMO−EHOMO

(2) A=−ELUMO

(3) I=−EHOMO

(4) χ=I+A2

(5) η=I−A

(6) ω=μ22η

(7) μ=−I+A2

(8) σ=1η

Results

Compound collection and ADMET prediction

A total of 190 compounds of Panax ginseng reported in the TCMSP database (Accessed on: 14 April 2023) were collected and optimized with LigPrep module. Then QikProp was used to screen 190 compounds that predict the ADME properties. On ADME screening, 27 compounds were noticed to fulfill most drug-likeness properties (File S1) and selected for molecular docking.

Differential expression analysis

Meanwhile, our search in NCBI, GEO datasets (Accessed on: 30 April 2023) allowed us to identify the suitable dataset GSE120774 based on the selection criteria. The GSE120774 dataset consists of 19 human epicardial adipose tissue (EAT) from ten controls and nine CAD that experimented in microarray platform GPL6244 ((HuGene-1_0-st) Affymetrix Human Gene 1.0 ST Array (transcript (gene) version)). By employing limma programming, a total of 3,499 DEGs (1,408 up and 2,091 down-regulated) with log2FC and p < 0.05 were identified. However, only the over-expressed 1,408 genes in CAD were selected for the subsequent assessment.

Protein–protein interaction and enrichment analysis

Using the data retrieved from EXOCARTA, 6,394 unique exosomal proteins were identified and mapped with the 1,408 over-expressed CAD genes. A total of 489 proteins identified to be common were termed as SET-A, as mentioned in the methodology section. Then, the protein network that was constructed with the 489 (SET-A) proteins was extended to have 1,487 (489+998) proteins to form a complex network (Fig. 1), suggesting that these 489 proteins on exosomal delivery to a target cell can readily interact with the 998 cellular proteins that might influences molecular function of the target cell. Thus, the sub-network was constructed to capture the proteins from SET-A that have a high degree of interaction with the proteins (998) of the extended network. Notably, Mothers against decapentaplegic homolog 2 (SMAD2) from the SET-A showed high degree of interaction in the protein network (Fig. 2). Consequently, the gene enrichment analysis was carried out with 1,487 proteins. The top 20 enriched GO terms were shown in Fig. S1. Likewise, the top 20 enriched molecular pathways were illustrated in Fig. S2. On the assessment of sub-network (Fig. 2), SMAD2 protein was noticed with the highest interacting proteins (n = 174) that represented as putative target for molecular docking.

Figure 1 Protein interaction network.

Protein–protein interaction network constructed using the STRING interaction network tool for the proteins that are common between exosomal cargo and CAD over-expressed genes. The colored nodes represent the proteins and the line denotes the edges/connectivity between the proteins. The black node denotes the highly connected SMAD2 protein in the network.

Figure 2 SMAD2 Sub-network.

The crucial sub-network consisting of highly connected nodes extracted from theprimary network. This extracted networkshows SMAD2 (yellow node) has the highest connectivitywith other proteins.

Molecular docking with SMAD2

The protein structure of SMAD2 was retrieved from PDB (ID: 5XOD) and optimized to perform a docking with 27 natural compounds along with five references CAD drugs (amlodipine, furosemide, digoxin, clopidogrel, and atorvastatin). The active site for the SMAD2 was determined as GLU270, PRO271, PHE273, TRP274, TYR340, GLY342, GLY343, VAL345, PHE385, ASN387, LEU442, GLY444, and PRO445 using P2Rank tool. The compound inermin (MOL003648) has the highest binding score (−5.024 kcal/mol). Likewise, furosemide showed highest affinity of −4.472 kcal/mol with SMAD2 compared other analyzed CAD drugs. These results indicate that the compound from Panax ginsenghas a higher affinity towards SMAD2 than the CAD drugs. Top 10 molecular docking results were represented in Table 1 and the interaction of the inermin and furosemide with SMAD2 were displayed as Figs. 3A & 3B. Overall docking results of 27 natural compounds and five references CAD drugs were given in the File S2.

Table 1 Top 10 docking score of 27 active compounds of Panax ginseng along with five CAD drugs against SMAD2.

Compounds	Docking score
(kcal/mol)	
MOL003648	−5.024	
MOL004100	−5.014	
MOL005384	−4.702	
MOL005321	−4.657	
MOL005314	−9.32	
Furosemide (CID-3440)	−4.472	
MOL002136	−4.308	
MOL005308	−4.194	
MOL005369	−4.091	
Digoxin (CID-2724385)	−3.971	

Figure 3 Protein–ligand interaction.

The Ligand Interaction module in Maestro, Schrödinger was used to display the ligand-protein interaction. Molecular interaction of inermin (A) and furosemide (B) with SMAD2. Inermin forms a more favorable interaction than the known drug (furosemide).

Molecular dynamic simulation of SMAD2—inermin complex

The docked Panax ginseng compound inermin and the known CAD drug furosemide with SMAD2 protein were used for MD simulation. The RMSD plot shows the inermin-SMAD2 and furosemide-SMAD2 complexes were tend to deviate initially and stabilized at the end of the simulation (Figs. 4A & 4B). The average RMSD value of the inermin-SMAD2 complex was 2.67 Åand the average of furosemide-SMAD2 complex was 3.69 Å. From the RMSD trajectory, the inermin has value less than 3 Å, which indicates inermin-SMAD2 formed a stable complex. Based on the RMSF plots, the inermin-SMAD2 complex (Fig. 5A) shows lower fluctuation than the furosemide -SMAD2 complex (Fig. 5B) during the 100 ns MDS run. Inermin (Figs. 6A & 7A) and furosemide (Figs. 6B & 7B) had 35 and 22 ligand contacts with the SMAD2 target. Further, the inermin average values of Rg, SASA and molecular surface area were 3.44 Å, 258.7 Å2, and 247.75 Å2, respectively (File S3) (Fig. 8A). Likewise, the average Rg, SASA, and molecular surface area trajectories for furosemide properties were 3.82 Å, 211.91 Å2, and 269.55 Å2 (Fig. 8B) (File S3).

Figure 4 The root-mean-square-deviation (RMSD) of (A) inermin and (B) furosemide with SMAD2 protein.

The RMSD trajectories were obtained by using the simulation interaction diagram module in Maestro, Schrödinger. The blue color trajectory indicates protein (C α) RMSD and the red trajectory indicates ligand fit protein RMSD.

Figure 5 The protein root mean square fluctuation of inermin (A) and furosemide (B) with SMAD2 complex.

The simulation interaction diagram module in Maestro Schrödinger was utilized to assess the RMSF trajectories of inermin-SMAD2 (A) and furosemide-SMAD2 (B) complexes. The blue trajectory indicates protein (C α) residual fluctuation during 100 ns simulation.

Figure 6 Protein-ligand contact of inermin (A) and furosemide (B) with SMAD2.

Protein-ligand contacts were obtained from the simulation interaction diagram module in Maestro, Schrödinger. Four levels of ligand-protein contacts 1. Hydrogen bond (Green), 2. Ionic bond (Pink), 3. Hydrophobic bond (Purple), and 4. Water bridges (Blue) were noticed during simulation. The contact plot showed inermin (A) with a higher number of contacts with the SMAD2 protein during the course of simulation than the furosemide (B).

Figure 7 Timeline of protein-ligand contacts during 100 ns simulation (A) inermin-SMAD2 complex and (B) furosemide-SMAD2 complex.

Protein-ligand contact timelines were obtained using the simulation interaction diagram module (Maestro, Schrödinger). Figure depicted the number of contacts formed between SMAD2 protein and ligands ((A): inerminand (B): furosemide) during the course of the simulation.

Figure 8 Ligand properties of inermin (A) and furosemide (B).

The simulation interaction diagram (Maestro, Schrödinger) was utilized to obtain ligand properties. The ligand properties include the root-mean-square-fluctuation (RMSF), radius of gyration (rGyr), intra hydrogen bond (intraHB), molecular surface area (MolSA), solvent accessible surface area (SASA), and polar surface area (PSA) during 100 ns molecular dynamics simulation. The ligand properties were used to analyze the changes in the ligand with SMAD2 protein. (A) The deviation of interimproperties throughout 100 ns simulation. (B) The deviation of furosemide properties for the period of 100 ns.

DFT quantum analysis—molecular geometry parameters

The optimized structure of inermin was shown in Fig. 9A. The bond length shows the distance between the nuclei of one atom and the other atoms. The angle between the two atomic bonds was determined by the bond angle analysis. The highest distance between the nuclei was noted in C10-C11 (1.56 Å), while O21-H33 (0.95 Å) has the shortest distance. The carbon-hydrogen bond has the lowest bond length compared to other atoms of inermin. In contrast, the carbon–carbon bond has the highest bond length, which shows that carbon interaction withholds high binding capacity. The lowest and the highest bond angle of inermin were observed between O21-H33 (0.95°) and C10 -C11 (1.56°). The lower torsion angle noted in C1-C2-C9-H25 (−179.91°), while greatest torsion angle observed in C15-C17-C18-O21 (179.94°). DFT analysis output data was provided in the File S4.

Figure 9 DFT analysis of inermin. (A) optimized structure, (B) HOMO and LUMO conformation, (C) electrostatic potential, and (D) Mulliken charges.

Optimized structure of inermin (A) was used to obtain highest occupied molecular orbital (HOMO), lowest unoccupied molecular orbital (LUMO), electrostatic potential, and Mulliken charges by using the Jaguar suite in Schrödinger. (B) The energy gap shows energy required to reach the LUMO from the HOMO of inermin. (C) The inermin electrostatic potential color represents positive (blue), negative (red), and neutral (white) regions. (D) Mulliken charges depicts the atomic charges of inermin.

DFT quantum analysis—HOMO and LUMO

The highest occupied molecular orbital (HOMO) and the lowest unoccupied molecular orbital (LUMO) are calculated with DFT-B3LYP6-311++ (dp). The frontier molecular orbital of inermin was illustrated in Fig. 9B. The HOMO orbital of inermin was partially covered without hydrogen atom, while the LUMO orbital fully dispersed except for the hydrogen atoms. The chemical activity of the molecule was calculated by correlating bond energies and their associated quantum parameters. The molecular behavior of the chemical compound was determined by the quantum parameters (ionization potential, electrophilicity (ω), electronegativity (χ), softness (S), hardness (η), and chemical potential (μ)). The electrophilicity scale shows that the atom has the ability to take up electrons. Likewise, the electronegativity scale describes the ability of the atom to attract electrons toward it. The hardness and softness values represent the electron charge transfer resistance and acceptance of an atom. The HOMO value was found to be −5.40 eV, while the LUMO was −0.45 eV, and has an energy gap (ΔE) of 4.95 eV (Fig. 9B). Furthermore, the calculated quantum parameters showed ionization potential (I) was 5.40 eV, electron affinity (A) was 0.45 eV, electronegativity (χ) was 2.93 eV, chemical hardness (η) was 4.95 eV, chemical potential (μ) was −2.93 eV, electrophilicity index (ω) was 1.46 eV, and chemical softness (σ) was 0.20 eV. Overall, the results suggest that inermin has high reactivity by quantum chemical analysis.

DFT quantum analysis—electrostatic potentials

The B3LYP/6-31G**++ in the Jaguar suite was utilized for ESP analysis. Inermin was analyzed to identify the reactivity, hydrogen bonding, and structural activity. The color variation denotes the electrostatic potential (Fig. 9C). Electrorophilic and nucleophilic attack regions were identified by the electrostatic potential. Particularly, blue, red, and white color indicates the positive, negative, and neutral electrostatic region. Interestingly (Fig. 9C), the major hydrogen atoms with high positive charge are prone to nucleophilic attack, which might be favorable for the protein-ligand interaction. The oxygen atoms present in inermin are negatively charged, which are expected to have nucleophilic attack. The higher electron density (deep red) was noted in the oxygen atom O21, which is likely to form a hydrogen bond with the protein (Fig. 9C).

DFT quantum analysis—Mulliken charges

The molecules capacity of the binding was determined by the electronic charges. The Mulliken charges of the representative atoms of inermin were shown in Fig. 9D. Surprisingly, all the hydrogen atoms are positively charged and the ones facing adjacent side to oxygen are highly positive compared to other hydrogen atoms. The electronegativity of the oxygen atom contribute strong interaction with the hydrogen atoms that ultimately supports the lead molecule to bind to protein efficiently (Fig. 9D). Likewise, the carbon atom in phenol ring interacted with oxygen atom that has a positive charge, while the others were negatively charged. The top three hydrogen atoms O21, O12, and O14 each had charge of −0.55, −0.52, and −0.51, which shows a negative value that might attract other atoms. In contrast, the C18 has the greater positive charge of 0.34 (Fig. 9D).

Discussion

CAD pathogenesis is the multifaceted events that includes oxidative stress, inflammatory response, cytokine activity, and cellular signaling (Kaulmann & Bohn, 2014; Stoll, Denning & Weintraub, 2006). Exosomes play an important role in cell–cell communication during the development of CAD (Gao et al., 2019). Evaluating the exosomal-associated protein would provide additional insights into CAD pathogenesis. Hence, we use a systems biological approach that integrates the exosomal proteins with the CAD over-expressed genes to identify the potential target for treating CAD. Despite recent advancements in treatment strategies, CAD continues to be a disease with high mortality world-wide. Additionally, a few of the currently available CAD drugs have significant adverse effects in the long term. Hence, investigating natural medicine that inhibits the vital disease progressing protein would reduce the adverse effects and improve the quality of life. Panax ginsengderived compounds have proven to have a multi-target potential, traditionally used to treat various diseases such as CAD, infertility, diabetes, and cancer (Liu et al., 2020). Previous research reports that Panax ginseng has excellent antioxidant and anti-inflammatory activity (Kim et al., 2017; Ahuja et al., 2018). Therefore, it is important to study the effect of Panax ginseng active compounds for the treatment of CAD.

In this study, we used the ADME prediction tool (QikProp) to find the active compounds from Panax ginseng by using various descriptors such as ADME-compliance score–drug-likeness parameters (range 0 to 5), the molecular weight of the molecule should range between 130.0 and 725.0, donor HB should range from 0.0 to 6.0, HB acceptor should range from 2.0 to 20.0, the partition coefficient of predicted octanol/water has to be ranged between −2.0 and 6.5, solubility range should be −6.5 to 0.5, cell permeability (Caco-2) (range: <25 is poor and >500 is great), predicted central nervous system activity (−2 denotes inactive and +2 denotes active), brain/blood partition coefficient between −3.0 to 1.2, MDCK cell permeability (range: <25 is poor and >500 is great), oral absorption (range: <25 is poor and >80% is high), human serum albumin binding has to range between −1.5 to 1.5, Lipinski’s rule of five violations (0–4), and Jorgensen’s rule of three violations (0–3) (Ntie-Kang et al., 2013; Rajagopal et al., 2020). Particularly, the Caco-2 assesses the bioavailability (cell permeability) of plant compounds in the human body (Ntie-Kang et al., 2013). The compounds which fulfilled these conditions were preceded by the next docking analysis. We have predicted that 27 major compounds of Panax ginseng have drug-likeness property. In addition, the potential protein target was identified by the systems biological approach that integrates an over represented gene in CAD that is actively transported by the exosome to other tissue leading to disease progression. Thereby, our analysis showed SMAD2 is the crucial protein that involved in the progression of CAD. SMAD2, known as SMAD Family Member 2, localized at human chromosome 18 encodes transcription factor protein. SMAD2 is linked with the critical pathways of CAD that include PI3K-Akt signaling, Ubiquitin mediated proteolysis, and focal adhesion pathway. More particularly, SMAD2 was linked with the pathogenic mechanism such as cellular damage and dysfunction (Sheikh et al., 2022), apoptosis (Marei et al., 2022), inflammation, and oxidative stress (Kotur-Stevuljevic et al., 2007). Furthermore, the whole exome sequencing demonstrated a connection between the pathogenic mutations of SMAD2 and phenotypes associated with cardiovascular illness, including arterial aneurysms and congenital abnormalities (Granadillo et al., 2018). Likewise, Zhang et al. (2017) observed a correlation between early-onset aortic aneurysms and the missense variant of SMAD2 (A278V). The risk of arterial aneurysmal disease may also be raised by the missense (S397Y, N361T, and S467L) and non-sense (N205*) variations of SMAD2 (Cannaerts et al., 2019). Furthermore, bicuspid aortic valve calcification may be partially promoted via the SMAD-dependent pathway (Zheng et al., 2021). Through the intracellular signaling pathway, the SMAD family functions as intracellular signaling effectors, modulating the transforming growth factor beta (TGF-b) family (Yuan & Jing, 2010). It is interesting to note that TGF-b is involved in many distinct cellular processes, including apoptosis, morphogenesis, migration, cellular homeostasis, proliferation, and growth (Yuan & Jing, 2010). Furthermore, according to Yuan & Jing (2010), the SMADs family is also linked to a number of cardiac conditions, including cardiac fibrosis, myocardial infarction, pulmonary artery hypertension, and cardiac failure. Interestingly, in our analysis SMAD2 was determined to be transported via exosome and interacts with multiple proteins in target cells on the delivery. Notably, our investigation identifies that SMAD2 can interact with proteins in the target cell that could change the cellular environment in the pathological state.

We performed molecular docking between SMAD2 with active compounds of Panax ginseng along with five standard drugs currently used for the treatment of CAD. SMAD2 protein used for docking identifies that inermin (−5.024 kcal/mol) could be effective than furosemide (−4.472 kcal/mol) based on binding score. Henceforth, inermin will be effective in inhibiting the protein and might reduce the progression of CAD via exosome transport (Fig. 3). In this context, we selected inermin and furosemide to perform MD simulation until 100 ns.

MD simulation is used for analyzing the stability and conformation changes in the protein-ligand complex (Sokkar et al., 2021). The molecular dynamic behavior of inermin and furosemide interaction with the binding sites of SMAD2 was studied through 100 ns simulation to provide additional insights. The RMSD trajectory analysis is used to analyze the stability of the protein-ligand complex (Sokkar et al., 2021). Our trajectory analysis showed SMAD2-inermin complex RMSD of protein Ca (alpha carbon), backbone, and heavy atoms was distinguished in the range of 1.18–3.47 Å, 1.22−3.46 Å, and 1.39–3.61 Å, respectively (Fig. 4A). Likewise, the SMAD2-furosemide complex RMSD of protein Ca, backbone, and heavy atoms was observed in the range of 1.33–4.81 Å, 1.36−4.76 Å, and 1.53–4.78 Å, respectively (Fig. 4B). The Ca atoms of SMAD2-inermin complex fluctuated in the range of 1.18–3.47 Å and finally stabilized after 75 ns of simulation. These results indicate that inermin-SMAD2 complex is equilibrated (Sivasubramanian et al., 2009). Similarly, the Ca atoms of the SMAD2-furosemide complex fluctuated in the 1.33–4.81 Å range and stabilized after 30 ns of simulation. In SMAD2-inermin complex, a higher fluctuations (up to 3.47 Å) in RMSD were observed at 50–55 ns and 92 ns. Likewise, the RMSD of SMAD2-furosemide complex expressed higher fluctuations (up to 4.81 Å) between 5–10 ns. During 100 ns of simulation, stable hydrophobic interactions were observed with ARG427 for SMAD2-inermin complex. While SMAD2-furosemide complex formed stable hydrogen between 25–100 ns simulation. The amino acid fluctuation on the bonding of compounds is examined by using RMSF trajectories (Fig. 5). SMAD2-inermin complex exhibited high flexibility with RMSF in residues ARG365 (Ca: 4.82 Å; backbone: 4.85 Å) and GLY260 (Ca: 6.38 Å; backbone: 6.00 Å), whereas lower RMSF was reported for residues SER417 (Ca: 0.44 Å; backbone: 0.46 Å) and VAL345 (Ca: 0.44 Å; backbone: 0.45 Å). Meanwhile, the SMAD2-furosemide complex has higher RMSF value in amino acids, GLY260 (Ca: 7.87 Å; backbone: 7.57 Å) and PRO261 (Ca: 7.18 Å; backbone: 7.24 Å), whereas low RMSF was reported for residues VAL345 (Ca: 0.51 Å; backbone: 0.51 Å) and PHE346 (Ca: 0.51 Å; backbone: 0.53 Å).

A total of 35 ligand contacts (Figs. 6 and 7) were formed with amino acids of protein, from SER30-LEU32, GLY34, GLU270-ALA272, TRP274, LEU282-GLN284, SER293, GLN294, PRO295, GLU309, ASN320, ASN322, THR324, TYR340, GLY342, GLN364, ARG365, PRO370, ALA371, LYS420, GLY423, ALA424-THR430, THR432, and SER433 between inermin and SMAD2. Likewise, there are 22 residual contacts were observed between SMAD2 and furosemide which include SER30-GLY34, GLY260-ASP262, GLU270-TRP274, SER293-PRO295, SER308-TYR340, GLY342, and GLY343. As depicted in Fig. 6, both compounds were stabilized by forming a salt bridge interaction (∼80%). SMAD2-inermin complex has higher hydrophobic interaction in ARG427 and ARG428. Similarly, furosemide formed a strong hydrogen bond with SMAD2 residue LEU32. The protein-ligand contacts timeline was represented in Fig. 7.

As represented in Figs. 3 and 7 the interaction of inermin and SMAD2 shows that SER293 sustained its docking pose and formed a hydrogen bond during the course of the simulation. Likewise, the furosemide interacted with SMAD2 and retained its docking pose with amino acids (ALA272, SER293, and TYR340) by forming a hydrogen bond. Inermin compound donated one hydrogen atom to the amino acid SER293. While furosemide donated two hydrogen bonds to the ALA272 and SER293 and accepted one hydrogen from TYR340. The RMSF value of the inermin interacted residue SER293 has 0.84 Å. The RMSF value of the furosemide interacted with ALA272, SER293, and TYR340 was 1.22, 0.96, and 0.58. Both compounds showed lower RMSF value in their respective binding sites, which shows lower flexibility. Inermin polar surface area ranged 117.42–130.41 Å2, Rg extended from 3.19 to 3.66 Å, SASA ranged from 153.77–474.58 Å2, molecular surface area ranged from 243.53–251.28 Å2. Similarly, the furosemide properties (Fig. 8) such as polar surface area, Rg, SASA, and molecular surface area of ligands were noted in the range of 208.75–242.62 Å2, 3.31−3.98 Å, 109.55–353.55 Å2 and 258.06–276.24 Å2, respectively. According to Fig. 8, the inermin has higher surface area than the furosemide, which shows that inermin has interacted with the solvent readily. Further the inermn has lower Rg value compared to the furosemide, which shows that the inermin-SMAD2 formed a compact structure during the course of simulation. Additionally, we performed DFT analysis to find the reactivity of inermin. Our analysis showed that inermin is highly reactive and might interact with the protein readily. Moreover, ESP analysis also showed that inermin has both nucleophilic and electrophilic attack region. Figure 9 depicts the increased electronegativity of the inermin acceptor’s hydrogen bond resulted in stronger hydrogen bond formation with the SMAD2 in accordance with docking. Although, this study demonstrates the potential utility of natural compounds for CAD, there are a few limitations that are need to be considered. Particularly, the results presented in this study are computationally obtained, further cell culture or animal model validation are required to acquire more reliable outcome. Additional studies are needed to identify the toxicity, dosage, and efficacy of the selected plant compound in a consistent manner.

Conclusions

The potential target SMAD2 was identified through systems biological analysis by integrating CAD-associated exosomal genes. Simultaneously, we identified 27 major lead compounds of Panax ginseng by ADME analysis. Molecular docking against SMAD2 determined that inermin has a higher binding score than the control drug. In addition, the MD simulation confirmed that the SMAD2 with inermin attained stability during the 100 ns MD run. Also, the quantum chemical properties of inermin showed high chemical reactivity. We conclude that the inermin will be a suitable alternative for the treatment of CAD. However, further validation was needed before proceeding with the clinical trial.

Supplemental Information

Supplemental Information 1 Supplementary Figures

Click here for additional data file.

Supplemental Information 2 ADME screening of Panax ginseng compounds

The drug-likeness properties of all 27 compounds

Click here for additional data file.

Supplemental Information 3 Molecular docking results of 27 natural compounds and five references CAD drugs

Click here for additional data file.

Supplemental Information 4 Moleular trajectories for the inermin and furosemide with SMAD2

Click here for additional data file.

Supplemental Information 5 Density functional theory

DFT analysis of inermin compound derived from Panax ginseng

Click here for additional data file.

All authors thank their Institutes for providing computational support to complete this research.

Additional Information and Declarations

Competing Interests

Author Contributions

Data Availability

The authors declare there are no competing interests.

Janakiraman V performed the experiments, prepared figures and/or tables, authored or reviewed drafts of the article, and approved the final draft.

Sudhan M performed the experiments, prepared figures and/or tables, and approved the final draft.

Abubakar Wani analyzed the data, prepared figures and/or tables, and approved the final draft.

Sheikh F. Ahmad analyzed the data, prepared figures and/or tables, and approved the final draft.

Ahmed Nadeem analyzed the data, prepared figures and/or tables, and approved the final draft.

Ashutosh Sharma conceived and designed the experiments, performed the experiments, prepared figures and/or tables, and approved the final draft.

Shiek S. S. J. Ahmed conceived and designed the experiments, performed the experiments, analyzed the data, prepared figures and/or tables, authored or reviewed drafts of the article, and approved the final draft.

The following information was supplied regarding data availability:

All result outcomes are available in the Supplementary Files.

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
