# Peer review of "Pharmacoscreening, molecular dynamics, and quantum mechanics of inermin from Panax ginseng: a crucial molecule inhibiting exosomal protein target associated with coronary artery disease progression"

_PeerJ, doi:10.7717/peerj.16481_

## Round 0.1 · original submission · Minor Revisions

Your manuscript entitled "Pharmacoscreening, molecular dynamics, and quantum mechanics of inermin from Panax ginseng: A crucial molecule inhibiting exosomal protein target associated with CAD progression", which you submitted has been reviewed.

The referee(s) would like to see some revisions made to your manuscript before publication. Therefore, I invite you to respond to the referee(s)' comments and revise your manuscript.

**Language Note:** The review process has identified that the English language must be improved. PeerJ can provide language editing services - please contact us at [email protected] for pricing (be sure to provide your manuscript number and title). Alternatively, you should make your own arrangements to improve the language quality and provide details in your response letter. – PeerJ Staff

Reviewer 1 ·

Basic reporting

English language corrections are needed.


In the abstract, the following sentence is misleading “Overall our combined analysis of target selection and phytochemical screening, inermin may act as a potential inhibitor against SMAD2 and helps manage CAD”. You could say “Overall, our computational study suggests that inermin could act against SMAD2 and may aid in the management of CAD” .


Scientific names should be in italics. Re-check the manuscript.


Follow similar identifiers. For instance, maintain uniformity with either “MOL003648” or “inermin”

Experimental design

No comment

Validity of the findings

No comments

Additional comments

Comments:
The authors designed novel frame work that integrates various computational methods such as ADME predictions, protein network analysis, molecular docking, molecular dynamics simulation, and DFT. Interestingly, this study identified exosomal putative targets by intersecting exosomal proteins and GEO datasets of CAD. Additionally, their approach identifies lead molecules from Panax ginseng that could lay a good foundation for to inhibit the progression of CAD.
The methodology is sound-full that are relevant to the objectives of the study. Conclusions are made from the outcome of the results derived from each step of the computational methods. However, some corrections are needed before publication. I recommend the manuscript for " revision". On justification and corrections of the given comments, the manuscript can be accepted for publication.


1. In the abstract, the following sentence is misleading “Overall our combined analysis of target selection and phytochemical screening, inermin may act as a potential inhibitor against SMAD2 and helps manage CAD”. You could say “Overall, our computational study suggests that inermin could act against SMAD2 and may aid in the management of CAD” .

2. Scientific names should be in italics. Re-check the manuscript.

3. Follow similar identifiers. For instance, maintain uniformity with either “MOL003648” or “inermin”.


4. Add limitations of the study at end the discussion section. More focus on experimental validation (as limitations) on future research.


5. Incomplete sentence “top over-expressed DEGs and the unique exosomal cargo proteins were identified (http://www.interactivenn.net/)”.


6. The target was predicted based on network analysis, so it should be expressed as a putative target.


“Additionally, SET-A SMAD2 protein with the highest interacting node degree to the 174 protein was selected as the target for molecular docking” .


7. In DFT quantum analysis - Mulliken charges, rephrase the following “The oxygen atom's electronegativity helps in the strong binding of hydrogen atoms.”

8. How do you relate DFT results with the docking results?

9. In discussion, the following sentence “MD simulation is used for analyzing the stability and conformation changes in the protein-ligand complex” needs citation.

10. Cite some article that uses PRANK for the prediction of binding pockets.

11. The following sentence needs to be reframed “Henceforth, inermin will be effective in inhibiting the protein and might reduce the progression of CAD via exosome transport (Fig. 5)”.


12. Unclear sentence “Higher fluctuations (up to 3.47 Å) in RMSD were observed at 50-55 ns and 92 ns in SMAD2-inermin complex and for SMAD2-furosemide complex higher fluctuations (up to 4.81 Å) were noted at 5-10 ns”.


13. In the conclusion section, the authors used a concrete statement like “inermin has a higher inhibitory effect than the control drug”, which could be changed to “higher binding score or higher binding affinity”.

Reviewer 2 ·

Basic reporting

The article " Pharmacoscreening, molecular dynamics, and quantum mechanics of inermin from Panax ginseng: A crucial molecule inhibiting exosomal protein target associated with CAD progression" by Janakiraman et al., describes screening of several molecules from a Chinese plant products using computational methods for effective inhibition of a protein associated with CAD progression. The authors have used neat and simple language throughout with proper referencing and ample background information. overall the paper is neatly structured with relevant results to the hypotheses stated.

Figures:
The article has figures with headings however, the legend or detailed description of figure is missing (image source, tools used for preparing the figure (figure 5), color codes etc.,)
A few other comments with figures:
Figure 1, Could the authors highlight SMAD2 in this big network?
Figure 3,4 Could the authors state the relevance of this presented data and the current manuscript. one suggestion is that these could be moved to supplementary figures instead.
Figure 6: inermin and furosemide are not labelled
Line 456 : Please use Figure.No of Fig. No uniformly throughout the manuscript
Figure 8,9: What programs were used to plot these figures?
Figure 11: Y axis label is unclear

References : Could the authors add references for the ADME properties stated (line 388 and Line 152)

Experimental design

The paper well fits the Aims and Scope of the journal and the experimental design is also met out well with sufficient details and information to replicate.

Few clarifications are requested for the following:
1. Line 145: what particular force field was used?
2. How long was the system equilibration carried on before MD production and what minimization algorithm was used?
3. Are there any other MD studies/ bioinformatic studies carried out earlier with the other referenced drugs taken for comparison? if yes, kindly mention

Validity of the findings

The authors have comeup with inerim as likely alternative drug candidate based on the results of their computational analysis subjected to further experimental evidences and limitations and further testing could be beneficial to draw definitive conclusions.

Additional comments

Line 152 - Caco2? what is this here, could you add a line explaining this and add a reference for the same?
Line 431- correct GLY_260 as GLY260

---

## Round 0.2 · Minor Revisions

The Section Editor, has commented and said:

"Some citations were missing and the authors should address how this research differs from these (and cite if appropriate):

Granadillo JL, Chung WK, Hecht L, Corsten-Janssen N, Wegner D, Nij Bijvank SWA, Toler TL, Pineda-Alvarez DE, Douglas G, Murphy JJ, Shimony J, Shinawi M. Variable cardiovascular phenotypes associated with SMAD2 pathogenic variants. Hum Mutat. 2018 Dec;39(12):1875-1884. doi: 10.1002/humu.23627.

Granadillo JL, Chung WK, Hecht L, Corsten-Janssen N, Wegner D, Nij Bijvank SWA, Toler TL, Pineda-Alvarez DE, Douglas G, Murphy JJ, Shimony J, Shinawi M. Variable cardiovascular phenotypes associated with SMAD2 pathogenic variants. Hum Mutat. 2018 Dec;39(12):1875-1884. doi: 10.1002/humu.23627.

Shi-Min Yuan, Hua Jing, Cardiac pathologies in relation to Smad-dependent pathways, Interactive CardioVascular and Thoracic Surgery, Volume 11, Issue 4, October 2010, Pages 455–460, https://doi.org/10.1510/icvts.2010.234773
https://www.ahajournals.org/doi/10.1161/circresaha.106.147264

There may be others, authors should do another literature search and cite appropriate, and distinguish or compare their research to other studies."

Please address this in a minor revision.

Reviewer 1 ·

Basic reporting

Clear and unambiguous, professional English used throughout.

Authors improved the manuscript, professional English used throughout. Henceforth, no further English language editing required.

Literature references, sufficient field background/context provided.

Professional article structure, figures, tables. Raw data shared.

Self-contained with relevant results to hypotheses.

Experimental design

Research question well defined, relevant & meaningful. Rigorous investigation performed to a high technical & ethical standard. Methods described with sufficient detail & information to replicate.

Validity of the findings

All underlying data have been provided; they are robust, statistically sound, & controlled.

Conclusions are well stated, linked to original research question & limited to supporting results

Additional comments

This study is intriguing and will advance our understanding of exosomal regulations in CAD. Notably, the authors have included each suggestion into the updated version of the revised manuscript. Henceforth, I endorse publishing this work in its current form in this journal.

---

## Round 0.3 · accepted · Accept

I have thoroughly reviewed the final version of your manuscript and can confirm that all comments and suggestions made by the reviewers have been adequately addressed. Although the original reviewers were not invited for this round of revisions, I have personally assessed the updated manuscript and am satisfied with the quality and content of the work presented.

Having gone through the document in detail, I can confidently state that this manuscript is ready for publication. Congratulations on this significant achievement, and I appreciate your dedication to producing high-quality work.